# Genomic determinants of Furin cleavage in diverse European SARS-related bat coronaviruses

Anna-Lena Sander[1], Andres Moreira-Soto [1], Stoian Yordanov[2], Ivan Toplak[3], Andrea Balboni [4], Ramón Seage Ameneiros[5], Victor Corman [1,6], Christian Drosten [1,6✉] & Jan Felix Drexler [1,6✉]

The furin cleavage site (FCS) in SARS-CoV-2 is unique within the *Severe acute respiratory syndrome–related coronavirus* (*SrC*) species. We re-assessed diverse *SrC* from European horseshoe bats and analyzed the spike-encoding genomic region harboring the FCS in SARS-CoV-2. We reveal molecular features in *SrC* such as purine richness and RNA secondary structures that resemble those required for FCS acquisition in avian influenza viruses. We discuss the potential acquisition of FCS through molecular mechanisms such as nucleotide substitution, insertion, or recombination, and show that a single nucleotide exchange in two European bat-associated *SrC* may suffice to enable furin cleavage. Furthermore, we show that FCS occurrence is variable in bat- and rodent-borne counterparts of human coronaviruses. Our results suggest that furin cleavage sites can be acquired in *SrC* via conserved molecular mechanisms known in other reservoir-bound RNA viruses and thus support a natural origin of SARS-CoV-2.

[1] Charité-Universitätsmedizin Berlin, Corporate Member of Freie Universität Berlin, Humboldt-Universität zu Berlin, Institute of Virology, Berlin, Germany. [2] Department of Zoology and Anthropology, Faculty of Biology, Sofia University "St. Kl. Ochridski", Sofia, Bulgaria. [3] Institute of Microbiology and Parasitology, Virology Unit, Veterinary Faculty, University of Ljubljana, Ljubljana, Slovenia. [4] Department of Veterinary Medical Sciences, Alma Mater Studiorum-University of Bologna, Via Tolara di Sopra 50, 40064 Ozzano Emilia, BO, Italy. [5] Group Morcegos de Galicia, Drosera Society, Pdo. Magdalena, G-2, 2° esq, 15320 As Pontes, Spain. [6] German Centre for Infection Research (DZIF), Associated Partner Charité-Universitätsmedizin Berlin, Berlin, Germany. ✉email: christian.drosten@charite.de; felix.drexler@charite.de

Emerging coronaviruses of recent or regular zoonotic origin include the betacoronaviruses Severe acute respiratory syndrome coronavirus (SARS-CoV), Middle East respiratory syndrome coronavirus (MERS-CoV) and Severe acute respiratory syndrome coronavirus 2 (SARS-CoV-2)[1]. SARS-CoV-2 is unique among emerging coronaviruses in its high transmissibility between humans[2]. SARS-CoV and SARS-CoV-2 belong to the species *Severe acute respiratory syndrome–related coronavirus (SrC)*, subgenus Sarbecovirus, and both use angiotensin-converting enzyme 2 (ACE2) as main cellular receptor[3]. In contrast to SARS-CoV, only SARS-CoV-2 contains a polybasic furin cleavage site (FCS) between the two subunits of the viral spike glycoprotein[4]. The FCS in SARS-CoV-2 is atypical in that it does not follow the canonical R-X-K/R-R amino acid residue pattern[5] but is defined by P/H-R-R-A-R with additional upstream basic residues occurring in some variants. The existence of the FCS has led to various hypotheses regarding the evolution of SARS-CoV-2, including conjectures about the possibility of a non-natural origin from laboratory experiments[6,7]. FCS naturally occur in different coronaviruses[8], such as MERS-CoV, which also harbors an atypical furin cleavage motif, namely P-R-S-V-R[9,10]. Efficient processing at the FCS is thought to be essential for entry into human lung cells and may also determine the efficiency of infection of the upper respiratory tract and consequent transmissibility of SARS-CoV-2[11]. So far, SARS-CoV-2 is unique among *SrC*, as even its closest known relatives, such as the bat coronaviruses RaTG13 from China and BANAL-20-52 from Laos, as well as the pangolin coronaviruses, lack an FCS[8,12,13]. The natural hosts of *SrC* are horseshoe bats, widely distributed in the Old World[14]. We and others previously showed that *SrC* in European horseshoe bats are conspecific with, but phylogenetically distinct from, *SrC* detected in Asia[15–18]. Here, we describe the S1/S2 genomic region encompassing the FCS in SARS-CoV-2 in ten unique European bat-associated *SrC* in comparison to other sarbecoviruses and mammalian coronaviruses. Our results provide evidence that furin cleavage sites can be natural acquired in the bat reservoir and thus support a natural origin of SARS-CoV-2.

## Results

**European bat SARS-related coronaviruses harbor remnants of furin cleavage sites.** We re-accessed stored original fecal samples from four horseshoe bat species (*Rhinolophus hipposideros, R. euryale, R. ferrumequinum, R. blasii*) collected in Italy, Bulgaria, Spain, and Slovenia during 2008–2009 and amplified a 816 base pair fragment of the viral *RNA-dependent RNA polymerase (RdRp)* of nine unique coronaviruses in addition to the bat-associated BG31 *SrC* previously described by us[17,18]. Taxonomic classification based on this fragment showed that all ten coronaviruses belonged to the species *SrC*, since translated amino acid sequence identity between the European bat *SrC* and SARS-CoV/SARS-CoV-2 was very high, ranging between 96.7–99.3% (Table 1). We then amplified and sequenced the partial or full

spike-encoding genomic region of European bat *SrC* containing the boundary between the S1/S2 spike subunits harboring the FCS in SARS-CoV/SARS-CoV-2. In a representative partial S2-based phylogeny that covered the genetic diversity of known *SrC*, European bat-associated *SrC* formed a sister clade to Chinese bat-associated *SrC* (Fig. 1, Supplementary Fig. 1). Sequence comparison of the S1/S2 genomic region revealed sequence motifs (R-X-R) that partially resemble the atypical polybasic FCS of SARS-CoV-2 at the S1/S2 boundary in 6 bat-associated *SrC* from Spain, Slovenia and Bulgaria (Fig. 2).

**Furin cleavage sites occur in animal homologues of human coronaviruses.** To better understand the genealogy of furin cleavage at the S1/S2 site within human coronaviruses (HCoVs), we investigated the genomic region encompassing potential FCS within human-associated coronaviruses and their animal homologues by using a furin cleavage prediction algorithm (ProP). Within animal homologues of bat-associated human coronaviruses, a furin cleavage motif in the S1/S2 genomic region only exists in MERS-related coronaviruses. Here, 10/27 (37%) of bat-associated and all camel-associated MERS-related viruses showed an FCS motif in the S1/S2 genomic region. Six out of those 10 bat-associated MERS-related viruses had a canonical R-X-K/R-R motif, whereas the remaining bat- as well as all camel- and human-derived sequences followed the R-X-X-R motif allowing furin cleavage in MERS-CoV[9]. These results suggest that an FCS may not be selected in bat hosts, as compared to viruses observed in humans or intermediate hosts (Fig. 3 and Supplementary Data 1). Within the two HCoVs likely originating from a rodent reservoir, HCoV-OC43 and HCoV-HKU1, a canonical R-X-K/R-R at S1/S2 site was present in all related animal CoVs, including all viruses known from the likely rodent reservoir, potentially hinting at differences in organ tropism and/or transmission routes between bat and rodent coronavirus reservoirs.

**Furin cleavage sites may be acquired through conserved molecular mechanisms in the bat reservoir.** The hypothetical turnover of FCS in the animal reservoir and fixation of FCS after host switches is reminiscent of the prototypic example of an RNA virus gaining pathogenicity via acquisition of an FCS, the avian Influenza A virus (AIV). AIVs are distinguished into low-pathogenic avian influenza virus (LPAI) and high-pathogenic avian influenza virus (HPAI). HPAIs are defined by the existence of a polybasic FCS in the hemagglutinin (HA) protein. A major determinant of the transition from LPAI into HPAI within the reservoir or the new host[19,20] is the acquisition of an FCS, which can be introduced through three different molecular mechanisms: recombination with cellular or other RNA molecules, multiple nucleotide insertions, or nucleotide substitutions[20–22]. Both insertions and nucleotide substitutions in the HA gene of AIVs are facilitated by a stem-loop secondary RNA structure enclosing the FCS and a high adenine/guanine content in the external loop structure[22]. Importantly, genomic

**Table 1 Amino acid identities of a partial *RdRp* fragment of European bat coronaviruses and prototype betacoronaviruses for species delineation.**

| % amino acid sequence identity (range): | | | | | |
|---|---|---|---|---|---|
| Within European bat CoVs | European bat CoVs compared to | | | | SARS-CoV[a] Vs. SARS-CoV-2[a] |
| | SARS-CoV[a] (AY274119) | SARS-CoV-2[a] (MT019529) | Hp-BetaCoV (NC025217) | MERS-CoV[a] (NC019843) | |
| 97.8–100% | 97.4–99.3 | 96.7–97.8 | 82.4–82.7 | 75.4–75.7 | 98.2 |

[a]For clarity of presentation, only one reference strain of SARS-CoV and SARS-CoV-2 as representatives of the species *SARS-related coronavirus*, Hp-Betacoronavirus as representative of the species *Bat Hp-betacoronavirus Zhejiang2013* and MERS-CoV as a representative of the species *Middle East respiratory syndrome-related coronavirus* was included in the analysis.

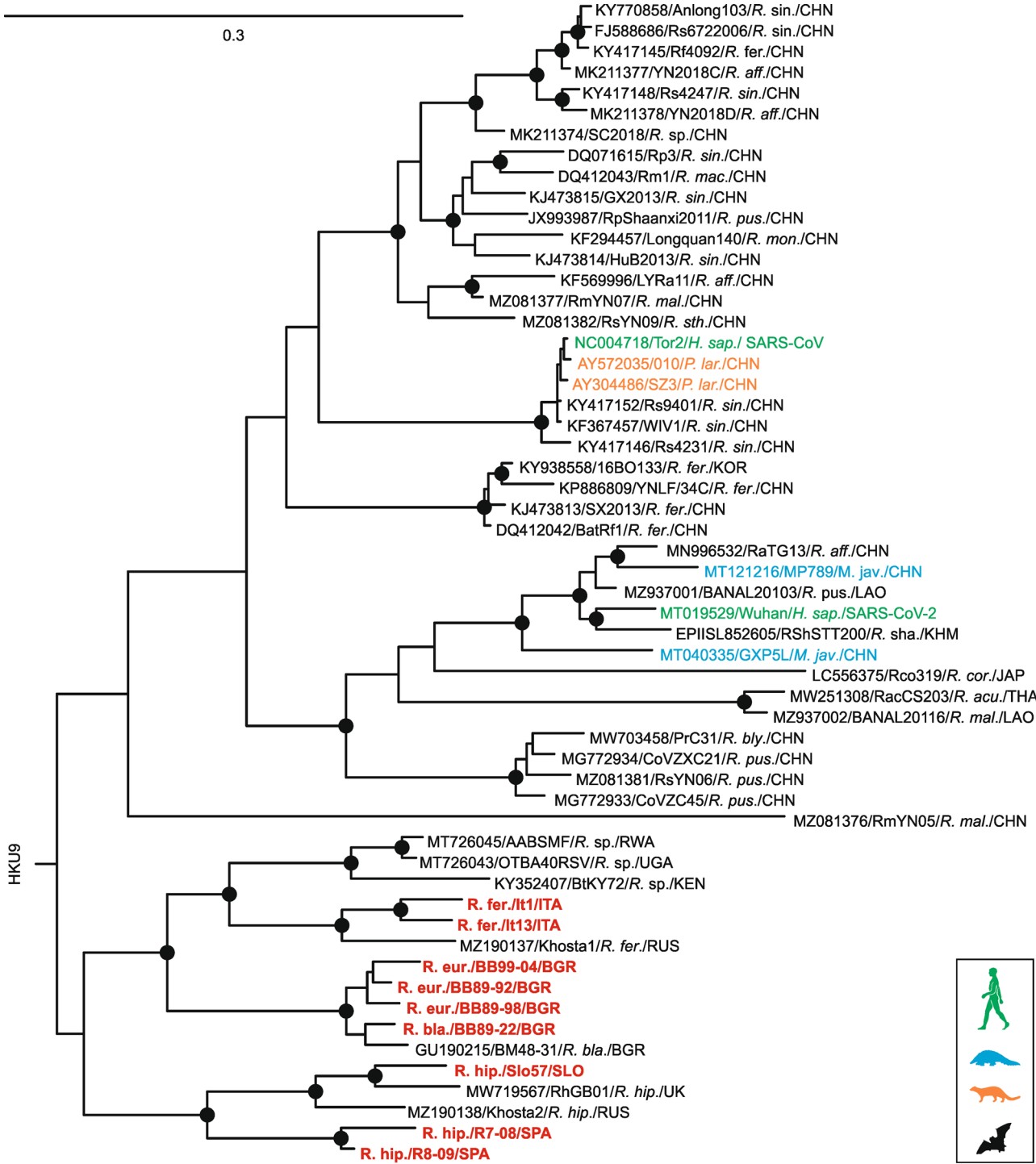

**Fig. 1 Evolutionary relationships of SARS–related coronaviruses.** Phylogeny based on a partial S2-fragment (495 nt) of representative sequences of the complete diversity of *SARS-related coronaviruses* (*SrC*) shown in Supplementary Fig. 1. European horseshoe bat *SrC* generated within this study are shown in bold red and without accession numbers. Sequences are named as followed: GenBank accession number/strain name/host species/country of detection. Circles at nodes indicate support of grouping in ≥90% of 1000 bootstrap replicates. Scale bar represents nucleotide substitutions per site. *H. sap.*, *Homo sapiens*; *M. jav.*, *Manis javanica*; *P. lar.*, *Paguma larvata*; *R. acu.*, *Rhinolophus acuminatus*; *R. aff.*, *Rhinolophus affinis*; *R. bla.*, *Rhinolophus blasii*; *R. bly.*, *Rhinolophus blythi*; *R. cor.*, *Rhinolophus cornutus*; *R. eur.*, *Rhinolophus euryale*; *R. fer.*, *Rhinolophus ferrumequinum*; *R. hip.*, *Rhinolophus hipposideros*; *R. mac.*, *Rhinolophus macrotis*; *R. mal.*, *Rhinolophus malayanus*; *R. mon.*, *Rhinolophus monoceros*; *R. pea.*, *Rhinolophus pearsonii*; *R. pus.*, *Rhinolophus pusillus*; *R. sha.*, *Rhinolophus shameli*; *R. sin.*, *Rhinolophus sinicus*; *R. sth.*, *Rhinolophus stheno*; BGR, Bulgaria; CHN, China; ITA, Italy; JPN, Japan; KEN, Kenya; KHM, Cambodia; KOR, Korea; LAO, Laos; RUS, Russia; RWA, Ruanda; SLO, Slovenia; SPA, Spain; THA, Thailand; UGA, Uganda; UK, United Kingdom.

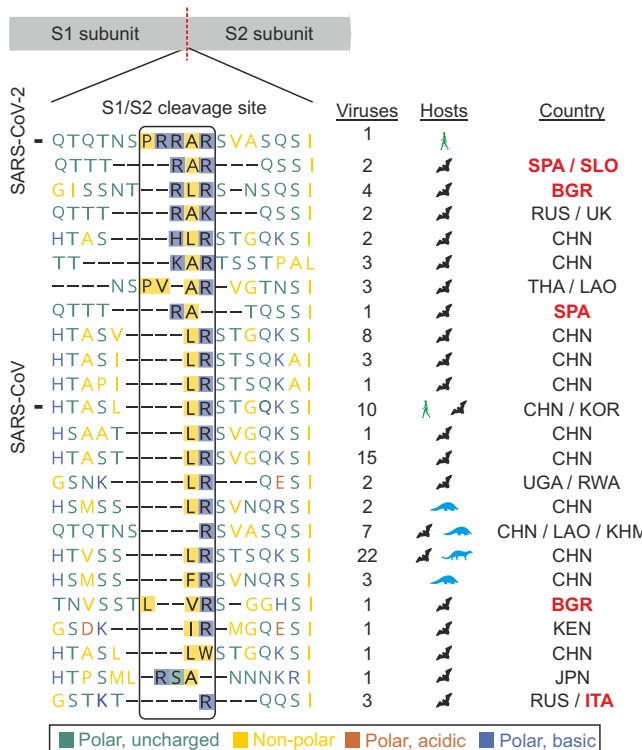

**Fig. 2 Comparison of the S1/S2 genomic region in SARS–related coronaviruses.** A scheme of the spike protein and its subunits S1 and S2 shows the position of the furin cleavage site (FCS) in SARS-CoV-2. Amino acid residues of the polybasic FCS in SARS-CoV-2 are highlighted within the box. European horseshoe bat *SrC* sequences generated within this study are indicated by country names in bold red. BGR, Bulgaria; CHN, China; ITA, Italy; JPN, Japan; KEN, Kenya; KHM, Cambodia; KOR, Korea; LAO, Laos; RUS, Russia; RWA, Rwanda; SLO, Slovenia; SPA, Spain; THA, Thailand; UGA, Uganda; UK, United Kingdom.

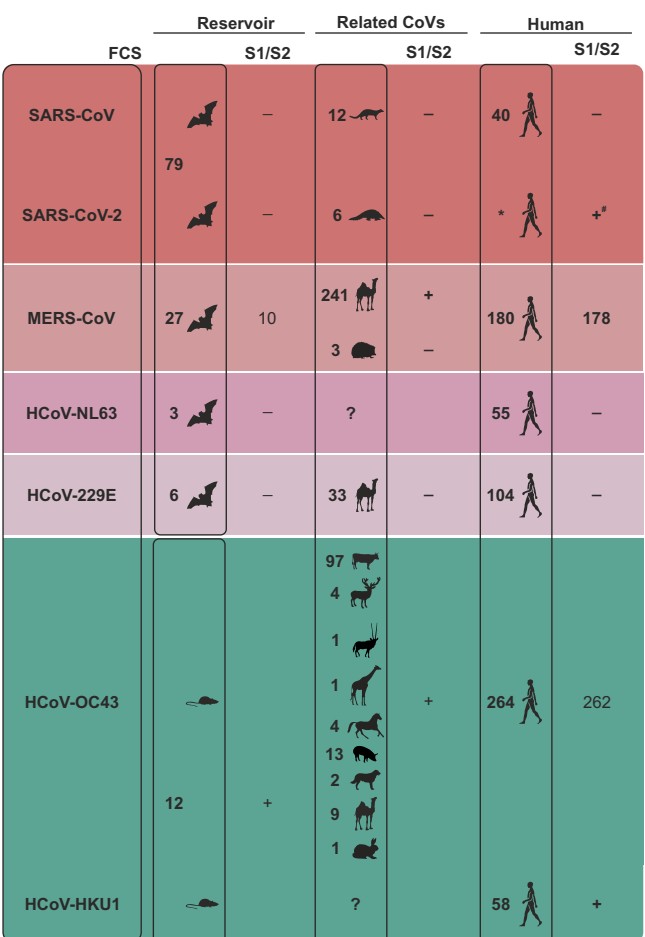

**Fig. 3 Furin cleavage sites at the S1/S2 interface in human and related animal coronaviruses.** Predicted furin cleavage sites with ProP scores are shown in the Supplementary Data 3. If not all tested sequences of one host category were predicted to contain an FCS, counts are given. *, 4,824,313 SARS-CoV-2 GISAID sequence entries; #304 SARS-CoV-2 sequences showing a loss of the RRAR furin cleavage site motif.

surrogates of all three mechanisms present in AIV are given in European bat-associated *SrC* which may enable acquisition of FCS (Fig. 4a).

First, within the region corresponding to the S1/S2 genomic region in SARS-CoV-2, some European bat-associated *SrC* are predicted to contain RNA secondary structures similar to HPAI sequences (Fig. 4b and Supplementary Figs. 2 and 3a). This result is in accordance with the generally high level of secondary structures within coronavirus genomes[23]. The existence of genomic premises prone for insertions or nucleotide substitutions suggests the feasibility of natural FCS acquisition in European bat-associated *SrC* similar to avian HPAI. Of note, a single non-synonymous nucleotide substitution (a C to G transversion) in the external loop of the RNA secondary structure would already suffice to create a motif resembling the SARS-CoV-2 R-R-X-R FCS in two European bat-associated *SrC* (T-R-L-R to R-R-L-R in BB99-04 and BB89-98) likely enabling furin cleavage based on in silico predictions (ProP cleavage site scores of 0.65 and 0.52 for BB99-04 and BB89-98, respectively; the program's threshold is 0.5) (Fig. 4a, b and Supplementary Fig. 3b). Deep sequencing of this genome position in BB99-04 and BB89-98 revealed presence of this transversion in 0.004% and 0.006% of the total reads in those two viruses, which was within the error rate of viral polymerases used for cDNA synthesis and amplification[24] (Table 2). Nevertheless, single nucleotide variants affording the emergence of furin cleavage in LPAI were found at a comparatively low frequency (0.0028%) in influenza virus strains, which may imply the potential for emergence of FCS in those

European bat *SrC* strains[22]. Whether bat *SrC* quasispecies harboring an FCS exist within European bat hosts thus requires careful examination.

Next, a more than 60% adenine/guanine content in the stem-loop structure of another European *SrC* (termed It1) may facilitate an insertion of an adenine/guanine stretch and thus the acquisition of a S1/S2 FCS motif (R-K-T-R; ProP cleavage site score of 0.69) (Fig. 4a, b and Supplementary Fig. 2a), comparable to the insertion leading to HPAI outbreaks in the US in 2016/2017[20,25].

Finally, the role of recombination allowing the acquisition of multibasic cleavage sites and the determinants thereof in AIV remain unclear[26], but may be facilitated by host nucleolar RNAs[27]. In coronaviruses, recombination is common[28] and thus represents an additional potential microevolutionary mechanism leading to acquisition of an FCS. As speculated for the origin of the S1/S2 FCS of SARS-CoV-2[29], recombination with other bat coronaviruses such as HKU9 could result in the acquisition of an FCS at the S1/S2 boundary. The existence of the palindromic sequence CAGAC in another European *SrC* (termed SLO57) comparable to SARS-CoV-2 (Fig. 4a) suggests that this genomic region may serve as an RNA signal for a recombination breakpoint in some European bat *SrC* leading to the acquisition of an S1/S2 FCS motif (R-Q-S-S to R-P-R-R-A-S-E-S-S including

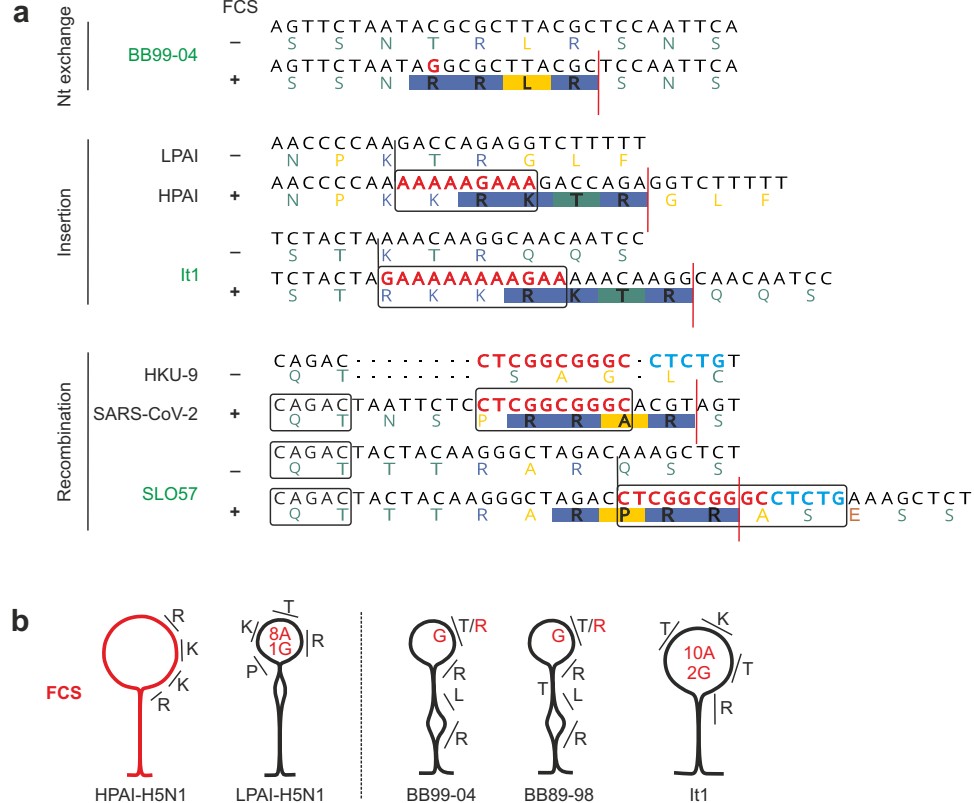

**Fig. 4 Predictive acquisition of furin cleavage sites in European bat SARS-related coronaviruses through different molecular mechanisms. a** Potential generation of FCS in different European bat *SrC* by nucleotide exchange, insertion due to external stem loop structures or recombination with HKU9 (15 nucleotides, 10 of which occur identically in SARS-CoV-2). Sequences of BB99-04 and It1 were trimmed in the figure for graphical reasons to highlight the FCS. The complete sequences used for FCS prediction were AKYGISSNRRLRSNSQSIV for BB99-04 and AKFGSTRKKRKTRQQSIL for It1. **b** Simplified predicted RNA secondary structures of the polybasic cleavage site regions of AIV and *SrC* that acquire FCS through nucleotide substitutions or insertions. Amino acids corresponding to codons forming the FCS are shown. Nucleotides and corresponding amino acids that make a change to an FCS through nucleotide substitutions or insertions are marked in red. Corresponding structures showing complete nucleotide sequences are shown in Supplementary Fig. 2.

**Table 2 Single nucleotide variants within two European *SrC* at spike position Thr672[a].**

| | | | | |
|---|---|---|---|---|
| BB99-04 | Consensus sequence | A | C | G |
| | Corresponding amino acid | Thr (T) | | |
| | A (%) | 1,46,506 (99.801) | 8 (0.005) | 159 (0.108) |
| | Corresponding amino acid | Thr (T) | Lys (K) | Thr (T) |
| | T (%) | 19 (0.013) | 187 (0.127) | 13 (0.009) |
| | Corresponding amino acid | Ser (S) | Met (M) | Thr (T) |
| | C (%) | 7 (0.005) | 1,47,250 (99.858) | 10 (0.007) |
| | Corresponding amino acid | Pro (P) | Thr (T) | Thr (T) |
| | G (%) | 259 (0.176) | 6[c] (0.004) | 1,46,817 (99.872) |
| | Corresponding amino acid | Ala (A) | Arg (R) | Thr (T) |
| | Total reads | 1,46,798 | 1,47,459 | 1,47,005 |
| BB89-98 | Consensus sequence | A | C | A |
| | Corresponding amino acid | Thr (T) | | |
| | A (%) | 1,17,032 (99.593) | 1 (0.001) | 1,16,893 (99.745) |
| | Corresponding amino acid | Thr (T) | Lys (K) | Thr (T) |
| | T (%) | 47 (0.0400) | 38 (0.032) | 13 (0.011) |
| | Corresponding amino acid | Ser (S) | Ile (I) | Thr (T) |
| | C (%) | 10 (0.009) | 117540 (99.961) | 8 (0.007) |
| | Corresponding amino acid | Pro (P) | Thr (T) | Thr (T) |
| | G (%) | 421 (0.358) | 7[bc] (0.006) | 278 (0.237) |
| | Corresponding amino acid | Ala (A) | Arg (R) | Thr (T) |
| | Total reads | 1,17,510 | 1,17,586 | 1,17,192 |

[a]Spike position Thr672 corresponds to the European *SrC* BB99-04 (GenBank Acc. No. KR559017).
[b]One of the reads was not paired-end.
[c]This singe nucleotide variant leads to a non-synonymous exchange generating an FCS motif in this virus.

the corresponding HKU9-derived sequence; ProP cleavage site score of 0.73)[29].

## Discussion

Our analysis suggests that natural acquisition of an FCS in European bat-associated *SrC* is conceivable. Even if the changes are hypothetical, they resemble prototypic molecular mechanisms leading to the generation of HPAI in their avian reservoir. It should be noted that in AIV, the acquisition of an FCS is one of several determinants in the transition from LPAI to HPAI, such as deletions in the neuraminidase or additional mutations in the HA protein[30]. It is therefore likely that additional genetic changes would have to occur before European bat-associated *SrC* could adapt to new hosts and acquire transmissibility via the respiratory tract. Cell entry via the host cell receptor ACE-2 and spike protein activation by TMPRSS2 via the S2' cleavage site are both essential for SARS-CoV, and SARS-CoV-2 pathogenesis[3,31–34]. The TMPRSS2 cleavage site KR|S in the S2' site seems to be highly conserved among bat *SrC*[11], indicating that the genomic prerequisites for cleavage by TMPRSS2 are given. Such traits of bat *SrC* increasing their zoonotic potential beyond FCS acquisition thus merit additional investigation.

In sum, our analysis presents several possible ways for natural acquisition of the FCS in SARS-CoV-2, supporting a natural evolutionary origin from bats with or without the involvement of intermediary hosts. Future studies of viral diversity in bats may identify other sarbecoviruses harboring functional FCS. The zoonotic potential of such sarbecoviruses deserves investigation to identify variants potentially posing threats to human health.

## Methods

**Sample collection and processing.** Bats (*R. euryale*, *R. blasii*, *R. ferrumequinum* and *R. hipposideros*) were sampled in four countries (Bulgaria, Italy, Slovenia, Spain)[15,16,18]. All animals were handled according to national and European legislation for the protection of animals (EU council directive 86/609/EEC). No animals were sacrificed during this study. All animal handling and sampling was done by trained personnel, with animal safety and comfort as the first priority during minimally invasive sampling (collection of faeces). Captured bats were freed from nets immediately and put into cotton bags for 2 to 15 min to allow them to calm down before examination. While being kept in bags, bats produced fecal pellets that were transferred to 500 µl RNAlater RNA stabilization solution (Qiagen, Hilden, Germany) for sample processing. Licenses for sampling of bats using mist nets, hand nets or harp traps were obtained from the respective countries and authorities: Bulgarian Ministry of Environment and Water, permit No. 192/26.03.200913, Italien Ministry of the Environment, permit No. 192/26.03.200952, Slovenian Environment Agency, permit No. 35701-80/200453, Service for the Biodiversity Conservation of the Rural Counseling of the Xunta de Galicia, Spain, permit No. 52/2010 n.s. 13697.

**Analysis of samples by reverse-transcription PCR.** Specimens were screened for the presence of coronaviral RNA by using nested reverse transcription-PCR (RT-PCR) amplifying 455 bp fragments of the *RNA-dependent RNA polymerase* (RdRp) gene. Briefly, RT-PCR was performed using the QIAGEN (Hilden, Germany) one-step RT-PCR kit with 5 µl of RNA, 200 nM of primer PC2S2 (equimolar mixture of TTATGGGTTGGGATTATC and TGATGGGATGGGACTATC), 900 nM of primer PC2As1 (equimolar mixture of TCATCACTCAGAATCATCA, TCATCA GAAAGAATCATCA, and TCGTCGGACAAGATCATCA) and 1 µl QIAGEN one-step RT-PCR kit enzyme mix. The amplification protocol comprised 30 min at 50 °C; 15 min at 95 °C; 10 cycles of 20 s at 94 °C, 30 s starting at 62 °C with a decrease of 1 °C per cycle, and 40 s at 72 °C; and 30 cycles of 20 s at 95 °C, 30 s at 52 °C, and 40 s at 72 °C[35]. For further phylogenetic analyses these amplicons were extended to >800 bp fragments towards the 5'-end using the 5' primer sequences: 5'-CTTCTTCTTTGCTCAGGATGGCAATGCTGC-3', SP3195, 5'-ATACTTT GATTGTTACGATGGTGGCTG-3'; SP3374, 5'-CTATAACTCAAATGAATCTT AAGTATGC-3', GrISP1, 5'-TTCTTTGCACAGAAGGGTGATGC-3'; and GrISP2, 5'-CTTTGCACAAAAAGGTGATGCWGC-3'[17]. One bat *SrC* spike glycoprotein gene (termed BB99-04; GenBank Acc No. KR559017) was fully characterized using nested RT-PCR primer sets (Supplementary Data 2).

S1/S2 genomic regions (346 bp; positions 23,422-23,767 in SARS-CoV-2 Wuhan strain GenBank Acc. Number MT019529) of eight bat *SrC* (GenBank Acc No. KC633198, KC633201, KC633203-205, KC633209, KC633212 and KC633217) were characterized using a hemi-nested RT-PCR assay using the following

oligonucleotides: panSARS-S1S2-F1 (TDGCTGTTGTHTAYCARGATGT), panSARS-S1S2-F2 (CARGATGTWAAYTGYACWGATGT) and panSARS-S1S2-R (CAGATGTACATDKTACAATCBAC). An extended S2 region (691 bp; positions 23,422-24,112 in SARS-CoV-2 Wuhan strain) for phylogenetic analyses was amplified using the same forward primers but a panSARS-S1S2-R2 (AGDCCATTRAACTTYTGHGCCACA) reverse primer. Briefly, RNA was reverse transcribed for 30 min at 50 °C using the SSIII One-Step Kit (Thermo Fisher) followed by 45 PCR cycles of 94 °C for 15 sec, 58 °C for 20 or 45 sec, respectively and 72 °C for 1 min. The 2nd round PCR was performed at the same conditions as the 1st round without reverse transcription. PCR amplicons were Sanger sequenced.

To detect single nucleotide variants within the S1/S2 site, PCR amplicons of the S1/S2 genomic region (346 bp fragments) were sequenced using the MiSeq System with the MiSeq Reagent Kit v2 (500-cycles) according to the manufacturer's instructions. Sequence reads obtained from the library were mapped against their corresponding S1/S2 genomic sequence in Geneious 9.1.8. A custom script using the python module pysam (https://github.com/pysam-developers/pysam) was utilized to determine the nucleotide distribution on specific genome positions from the read alignment.

**Phylogenetic analyses.** A tblastx search of the complete spike sequence of the Bulgarian *SrC* (GenBank Acc No. GU190215) within the taxonomy ID 11118 (*Coronaviridae*) excluding taxonomy ID 2697049 (SARS-CoV-2) was performed on 28 December 2021. Hits with percentage identities below 80% were non-*SrC* sequences and were thus not included in the dataset. SARS-CoV sequences from experimental infections or clones as well as sequences with less than 27,000 nt or gaps in the spike-encoding region were excluded, resulting in 88 sequences. One reference sequence of each SARS-CoV (NC004718) and SARS-CoV-2 (MT019529) as well as the nine sequences from European bats newly generated within this study were added, resulting in a final dataset of 99 sequences. Because coronaviruses frequently recombine[36] only the S2 region (495 nt) of the 691 nucleotide fragment was used for the phylogenetic analysis. Maximum-likelihood phylogenies were generated using FastTree[37] Version 2.1.10 using a GTR substitution model and 1000 bootstrap replicates. Local support values are based on the Shimodaira-Hasegawa (SH) test[38].

**In silico analyses.** Secondary structures were modeled using the UNAfold web server[39]. Furin cleavage sites were predicted using the ProP v.1.0b ProPeptide Cleavage Site Prediction software[10]. Sequences were retrieved from the NCBI Taxonomy website downloading all sequences of the species *Duvinacovirus* (HCoV-229E), *Merbecovirus* (MERS-CoV), *Setracovirus* (HCoV-NL63) and *Embecovirus* (HCoV-HKU1 and HCoV-OC43). Duplicates, non-complete genomes, sequences from experimental infections and clones were excluded from all datasets. Only the NCBI reference sequences of human genomes were left in the dataset at this stage of analysis. For Sarbecoviruses, the dataset of the tBlastx search used for the *SrC*-phylogeny of the Supplementary Fig. 1 was used. Translation alignments were conducted in Geneious 9.1.8. Accession numbers of all viruses tested in ProP are listed in the Supplementary Data 3. Amino acid identity in translated 816 nt *RdRp* fragments was calculated with MEGA11[40] using the pairwise deletion option.

**Data collection of HCoV sequences for examining conservation of FCS in human sequences.** Datasets used for the ProP FCS prediction included only one human reference sequence. However, conservation of FCS in human-derived sequences was examined as follows. Blastp searches of partial spike sequences flanking the S1/S2 site (SARS-CoV, NC004718, positions (pos.) 23,139–24,191; HCoV-229E, NC002645, pos. 21,923–22,954; HCoV-NL63, NC005831, pos. 22,329–23,414; HCoV-OC43, NC006213, pos. 25,524–26,675; HCoV-HKU1, NC006577, pos. 24,823–25,977 and MERS-CoV, NC019843, pos. 23,358–24,449) was performed using the *nr* database on 3 January 2022. To restrict results to human-derived sequences only, the blast search was restricted to the taxonomy IDs 111137 (HCoV-229E), 290028 (HCoV-HKU1), 1335626 (MERS-CoV), 277944 (HCoV-NL63) and 31631 (HCoV-OC43). In the case of SARS-CoV, the species *Severe acute respiratory syndrome-related coronavirus* (taxonomy ID 694009) was chosen and non-human viruses were excluded according to their taxonomy IDs (1283333, 2709072, 1283332, 442736, 349342, 349343, 347537, 347536, 349344, 1508227, 1415834, 1415851, 1415852, 1699360, 1699361, 285945, 698398, 1487703, 1503296, 1503299, 1503300, 1503301, 1503302, 1503303, 2042697, 2042698, 2697049). Blastp search resulted in 157, 60, 378, 102, 348, 94 hits for HCoV-229E, HCoV-HKU1, MERS-CoV, HCoV-NL63, HCoV-OC43 and SARS-CoV respectively. Full records were downloaded and manually curated in Geneious 9.1.8, excluding experimental infections, clones, genetically modified sequences as well as sequences only partially covering the reference sequence resulting in a final dataset encompassing 104, 58, 180, 55, 264 and 40 protein sequences for HCoV-229E, HCoV-HKU1, MERS-CoV, HCoV-NL63, HCoV-OC43 and SARS-CoV respectively. Proteins sequences were aligned in Geneious 9.1.8 using *Mafft*[41] with an auto algorithm and a BLOSUM62 scoring matrix. Conservation of the FCS was examined manually.

Due to the immense amount of SARS-CoV-2 available sequences, we downloaded the metadata of all available sequences in GISAID on 11 January 2022, 10:55 am CET. Only metadata of sequences fulfilling the complete and high coverage criteria in GISAID were used, resulting in a final dataset encompassing metadata of 4,824,313 sequence entries. A custom python script was used to extract GISAID sample IDs of sequences with complete or partial deletions within the SARS-CoV-2 FCS motif (amino acid positions/motif $_{682}$RRAR$_{685}$) or of sequences which had at least one amino acid substitution at position R682 or R685, which all likely result in a non-functional FCS. These criteria matched 304 SARS-CoV-2 sequences, which were all of human origin. Sequences were downloaded from the GISAID database and aligned to the Wuhan reference strain MT019529 in Geneious 9.1.8 using *Mafft*.

**Reporting summary**. Further information on research design is available in the Nature Research Reporting Summary linked to this article.

## Data availability

All nucleotide sequences generated within this study were submitted to GenBank under accession numbers KC633198, KC633201, KC633202, KC633203, KC633204, KC633205, KC633209, KC633212, KC633217 and KR559017. KR559017 represents the full spike gene of the Bulgarian *SrC* BB99-04. MiSeq sequence reads have been deposited in the European Nucleotide Archive (ENA) under the study accession code PRJEB51900 with the read file accession codes ERR9434412 and ERR9434413.

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

## Acknowledgements

This study was funded by the German Research Foundation (DFG) (DR 810/7-1, DR 772/10-2), as well as the German Ministry of Research (01KI1723A). We thank Mara Battilani of the department of Veterinary Medical Sciences from University of Bologna, Italy, Danijela Černe of the Institute of Microbiology and Parasitology from University of Ljubljana, Slovenia, Florian Gloza-Rausch of the Behavioral Ecology and Bioacostics Lab at Museum für Naturkunde, Berlin, Germany for providing sample material, Ben Wulf for bioinformatical assistance and Monika Eschbach-Bludau for technical assistance.

## Author contributions

J.F.D. and C.D. designed research. A.L.S. and A.M.S. performed research, analysed data and performed artwork. V.C. performed research. S.Y., I.T., A.B., R.S.A. provided sample material. J.F.D. and C.D. supervised study concept. A.L.S. wrote the first draft of the manuscript. J.F.D., A.L.S., A.M.S. and C.D. wrote the manuscript. All authors have read the final version of the manuscript and accepted publication.

## Funding

## Competing interests

The authors declare no competing interests.
