## [Peer Review File · Communications Biology]

Reviewers' comments:

Reviewer #1 (Remarks to the Author):

The article written AL Sander et al. aims at presenting through the sequence analysis of known and recently characterized sequences of SARS-related beta CoV possible hypothesis explaining the natural origin of the Furin Cleavage Site at the S1/S2 site in the Spike protein of SARS-CoV-2. Overall, the sequence information of the European bat CoV presented in the article are of interest and can contribute to the understanding on putative molecular mechanisms that yielded to the apparition of the FCS in SARS-CoV-2. However, the article is extremely short and the figure is so compact that the reading of the brief-note and the understanding of the concepts are very hard, and may even lead to misunderstandings, in a field of research that rather needs clarification. The authors should definitely clarify their message.

Major comments

-The authors include in their analysis the evolution of S2' maturation site. It is not clear to me what the contribution of this part is. Demonstrating that a CFS can emerge naturally ? if so, this part should properly introduced and illustrated, showing how the CFS appeared (Insertion, deletion, mutations) and show this illustration contributes to the understanding of the mechanisms of the emergence of the CFS at S1S2 (non-functional insertion following by functional maturation through mutation? Or just mutation? Through what possible mechanism?). The biological role of the maturation on the Spike protein in S1/2 and S2', especially in interaction with the cellular receptors and fusion should be further presented for clarification.

- In the article, the authors compare the evolution of the CFS with the one observed in influenza. This could be misleading as the events leading to the CFS are likely not supported by the same mechanisms. For instance, recombination in CoV is mainly driven by template switching, which is not the case for influenza viruses. Is there any evidence that FCS of SARS-CoV-2 at S1/S2 is present elsewhere in the genome? Or in any other genome of beta CoV and that the recombination could have occurred in bats or intermediate hosts? For influenza virus, acquisition of polybasic sites can be explained by the presence of secondary structures involved in the slippage of the polymerase to increase the size of the cleavage site. I could not find any article describing such a mechanism for CoV. If existing it should be clarified to conclude if the example of It1 is relevant or not.

-If I understood correctly, in two of the original sequences, BB99-04 and BB89-98, a single mutation would be sufficient to create a CFS. Could the authors clarify if the coding sequence leading to the RRLR (or at least RRL) correspond to an extra sequence compared to other SrC, showing that they trapped "an evolution intermediate". If it is not the case, how these sequences support that the insertion can occur naturally? Of note, the sequences of interest should be presented in a table or an alignment and be accessible directly.

-Experimental procedures for sequencing and analysis of the quasispecies should be explicitly described. The translation of the products resulting from the nucleotide substitution should be presented in a figure of the main text.

-Conclusion : Whereas the work does support the conclusion that the S1S2 in some European SrC has predisposition to acquire FCS, the study, as currently presented, does not demonstrate the natural origin of SARS-CoV-2, and mechanisms might be different to the ones for influenza.

Minor Comments

L85: A>G is for me a transition and not a transversion.

Sup Table 2: replace S2 by S2' in the table, for clarification

Reviewer #2 (Remarks to the Author):

The authors analyzed existence of furin cleavage sites at S1/S2 and S2' in coronaviruses related to human diseases. Especially, viruses from bats (the main hosts of coronaviruses) and intermediate hosts are included. This provides a more complete picture of the origin of furin cleavages sites in SARs-CoV-2. The study helps us to understand the evolution of viruses causing human diseases.

Reviewer #3 (Remarks to the Author):

The authors re-assessed diverse SrC from European horseshoe bats (samples were collected during 2008-2009) and showed that these bat SrC are likely to evolve a furin cleavage site through three possible mechanisms: recombination, multiple nucleotide insertions and nucleotide substitutions. The latter two mechanisms can be facilitated by a stem-loop secondary RNA structure enclosing the furin cleavage site and a high purine content in the external loop structure which are exactly the cases in some European bat-associated SrC (such as It1, SLO57 and BB99-04). In addition, SLO57 has the palindromic sequence CAGAC in the S1/S2 region which suggests that this genomic region may serve as an RNA signal for a recombination breakpoint.

In summary, I think this concise paper offered new evidences that support existing three mechanisms that favor a natural gain of the furin cleavage site in SARS-CoV-2.

The paper is well written and the figure is really nice. I fully support its publication without change.

Reviewer #4 (Remarks to the Author):

I think this is an interesting analysis of the potential route that SARS-CoV-2 could have used to acquire the furin cleavage site (FCS). My main question is why? That's a tricky one to answer for sure but I think the authors should address this issue in their discussion. What evolutionary pressure is driving acquisition of the FCS? Allied to this they should discuss how this FCS is lost in cell culture (i.e. on passage in VeroE6 cells). I think this will round off a useful analysis.

Rebuttal letter for COMMSBIO-21-2594-T “Genomic determinants of Furin cleavage in diverse European SARS-related bat coronaviruses”

Dear Editors,

We thank our reviewers for their time and insightful comments. In response, we have carefully revised the manuscript. Please find a point-to-point reply to all comments below (identified in blue):

Reviewer 1:

The article written AL Sander et al. aims at presenting through the sequence analysis of known and recently characterized sequences of SARS-related beta CoV possible hypothesis explaining the natural origin of the Furin Cleavage Site at the S1/S2 site in the Spike protein of SARS-CoV-2. Overall, the sequence information of the European bat CoV presented in the article are of interest and can contribute to the understanding on putative molecular mechanisms that yielded to the apparition of the FCS in SARS-CoV-2. However, the article is extremely short and the figure is so compact that the reading of the brief-note and the understanding of the concepts are very hard, and may even lead to misunderstandings, in a field of research that rather needs clarification. The authors should definitely clarify their message.

Reply: We agree. The paper was submitted originally to another journal in a concise format and automatically forwarded. We have now re-formatted the manuscript according to the journal’s guidelines and separated what previously was a single figure into four main figures.

(end of reply)

Major comments:

1. The authors include in their analysis the evolution of S2’ maturation site. It is not clear to me what the contribution of this part is. Demonstrating that a CFS can emerge naturally ? if so, this part should properly introduced and illustrated, showing how the CFS appeared (Insertion, deletion, mutations) and show this illustration contributes to the understanding of the mechanisms of the emergence of the CFS at S1S2 (non-functional insertion following by functional maturation through mutation? Or just mutation? Through what possible mechanism?). The biological role of the maturation on the Spike protein in S1/2 and S2’, especially in interaction with the cellular receptors and fusion should be further presented for clarification.

Reply: We agree that the inclusion of the S2’ site was not helpful, since only cleavage at the S1/S2 site is relevant for the genealogy of SARS-CoV-2 and the claims regarding a putative non-natural origin. We have removed all the manuscript parts on S2’ and revised the complete results section and figures accordingly.

(end of reply)

2. In the article, the authors compare the evolution of the FCS with the one observed in influenza. This could be misleading as the events leading to the FCS are likely not supported by the same mechanisms. For instance, recombination in CoV is mainly driven by template switching, which is not the case for influenza viruses. Is there any evidence that FCS of SARS-CoV-2 at S1/S2 is present elsewhere in the genome? Or in any other genome of beta CoV and that the recombination could have occurred in bats or intermediate hosts? For influenza virus, acquisition of polybasic sites can be explained by the presence of secondary structures involved in the slippage of the polymerase to increase the size of the cleavage site. I could not find any article describing such a mechanism for CoV. If existing it should be clarified to conclude if the example of It1 is relevant or not.

Reply:

We agree with the reviewer that there are differences in the mechanisms used by coronaviruses and influenza A viruses (IAV). We revised the text accordingly and included new citations on the RNA secondary structures in SARS-CoV-2 (Simmonds, 2020) and scarcity of recombination in Influenza A viruses (Boni et al., 2008) (see below, new parts underlined).

Results section:

“Importantly, genomic surrogates of all three mechanisms present in IAV are given in European bat-associated *SrC* which may enable acquisition of FCS (Figure 4A).

First, within the S1/S2 genomic region in SARS-CoV-2, some European bat-associated *SrC* are predicted to contain RNA secondary structures similar to HPAI sequences (Figure 4B and Supplementary Figure 2A). This result is in accordance with the generally high level of secondary structures within coronaviruses genomes (29). The existence of genomic premises prone for insertions or nucleotide substitutions suggests the feasibility of natural FCS acquisition in European bat-associated *SrC* similar to avian HPAI. Of note, a single non-synonymous nucleotide substitution (a C to G transversion) in the external loop of the RNA secondary structure would already suffice to create a motif resembling the SARS-CoV-2 R-R-X-R FCS in two European bat-associated *SrC* (T-R-L-R to R-R-L-R in BB99-04 and BB89-98) likely enabling furin cleavage based on in silico predictions (ProP cleavage site scores of 0.65 and 0.52 for BB99-04 and BB89-98, respectively; the program’s threshold is 0.5) (Figure 4A and B and Supplementary Figure 2B). ...

“Finally, recombination remains a relatively rare event in IAV (32), but is a common in coronaviruses (33) and thus represents an additional potential microevolutionary mechanism leading to acquisition of a FCS in coronaviruses. As speculated for the origin of the S1/S2 FCS of SARS-CoV-2 (34), recombination with other bat coronaviruses such as HKU9 could result in the acquisition of a S1/S2 FCS. ... ”

Discussion section:

“Our analysis suggests that natural acquisition of an FCS in European bat-associated *SrC* is conceivable. Even if the changes are hypothetical, they resemble prototypic molecular mechanisms leading to the generation of HPAI in their avian reservoir. It should be noted that in AIV, the acquisition of a FCS is one of several determinants such as deletions in the transition from LPAI to HPAI, such as deletions in the neuraminidase or additional mutations in the HA protein (35). It is therefore likely that additional genetic changes would have to occur before European bat-associated *SrC* could adapt to new hosts and acquire transmissibility via the respiratory tract. Cell entry via the host cell receptor ACE-2 and spike protein activation by TMPRSS2 via the S2’ cleavage site are both essential for SARS-CoV, MERS-CoV and SARS-CoV-2 pathogenesis (3, 36-39). The TMPRSS2 cleavage site KRIS in the S2’ site seems to be highly conserved among bat *SrC* (10), indicating that the genomic prerequisites for cleavage by TMPRSS2 are given. Such traits of bat *SrC* increasing their zoonotic potential beyond FCS acquisition thus merit additional investigation.

In sum, our analysis presents several possible ways for natural acquisition of the FCS in SARS-CoV-2 (40), supporting a natural evolutionary origin from bats with or without the involvement of intermediary hosts. Future studies of viral diversity in bats may identify other sarbecoviruses harboring functional FCS. The zoonotic potential of such sarbecoviruses deserves investigation to identify variants potentially posing threats to human health.”

(end of reply)

3. If I understood correctly, in two of the original sequences, BB99-04 and BB89-98, a single mutation would be sufficient to create a CFS. Could the authors clarify if the coding sequence leading to the RLLR (or at least RRL) correspond to an extra sequence compared to other *SrC*, showing that they trapped “an evolution intermediate”. If it is not the case, how these sequences support that the insertion can occur naturally? Of note, the sequences of interest should be presented in a table or an alignment and be accessible directly.

Reply: The sequences of these two European bat SARS-related coronaviruses represent two viruses within the *SrC* gene pool in the bat reservoir. We show that the structural premises facilitating mutations (insertion or nucleotide exchange) are comparable to those found in avian influenza viruses. We detected these exact single nucleotide variants with a very low frequency by NGS but cannot prove their existence in nature. Please note that the sequence of BB99-04 is shown in Figure panel 3A. In response to your comment, we incorporated the figure element showing the C to G transversion and the resulting FCS acquisition for sequence BB89-98 in Supplementary Figure 2.

(end of reply)

4. Experimental procedures for sequencing and analysis of the quasispecies should be explicitly described. The translation of the products resulting from the nucleotide substitution should be presented in a figure of the main text.

Reply: Done as suggested. Please find the amino acid sequences resulting from the nucleotide substitution in Figure 4, as well as in the results section (see below).

“Of note, a single non-synonymous nucleotide substitution (a C to G transversion) in the external loop of the RNA secondary structure would already suffice to create a motif resembling the SARS-CoV-2 R-R-X-R FCS in two European bat-associated *SrC* (T-R-L-R to R-R-L-R in BB99-04 and BB89-98) likely enabling furin cleavage based on *in silico* predictions (ProP cleavage site scores of 0.65 and 0.52 for BB99-04 and BB89-98, respectively; the program’s threshold is 0.5) (**Figure 4A and B** and **Supplementary Figure 2B**).”

Furthermore, we incorporated the corresponding amino acid residues in a main text Table.

Table 2. Single nucleotide variants within two European *SrC* at spike position Thr672[§].

BB99-04	Consensus sequence	A	C	G
	Corresponding amino acid	Thr (T)		
	A (%)	146506 (99.801)	8 (0.005)	159 (0.108)
	Corresponding amino acid	Thr (T)	Lys (K)	Thr (T)
	T (%)	19 (0.013)	187 (0.127)	13 (0.009)
	Corresponding amino acid	Ser (S)	Met (M)	Thr (T)
	C (%)	7 (0.005)	147250 (99.858)	10 (0.007)
	Corresponding amino acid	Pro (P)	Thr (T)	Thr (T)
	G (%)	259 (0.176)	6* (0.004)	146817 (99.872)
	Corresponding amino acid	Ala (A)	Arg (R)	Thr (T)
Total reads	146798	147459	147005	
BB89-98	Consensus sequence	A	C	A
	Corresponding amino acid	Thr (T)		
	A (%)	117032 (99.593)	1 (0.001)	116893 (99.745)

Corresponding amino acid	Thr (T)	Lys (K)	Thr (T)
T (%)	47 (0.0400)	38 (0.032)	13 (0.011)
Corresponding amino acid	Ser (S)	Ile (I)	Thr (T)
C (%)	10 (0.009)	117540 (99.961)	8 (0.007)
Corresponding amino acid	Pro (P)	Thr (T)	Thr (T)
G (%)	421 (0.358)	7[#] (0.006)	278 (0.237)
Corresponding amino acid	Ala (A)	Arg (R)	Thr (T)
Total reads	117510	117586	117192

§Spike position Thr672 corresponds to the European *SrC* BB99-04 published under Accession number KR559017.

#One of the reads was not paired-end

*This single nucleotide variant leads to a non-synonymous exchange generating a FCS motif in this virus (end of reply)

Minor comments:

1. L85: A>G is for me a transition and not a transversion.

Reply: We apologize for this mistake. Corrected as suggested.

(end of reply)

2. Sup Table 2: replace S2 by S2' in the table, for clarification

Reply: We excluded the S2' site from our analyses and adapted the table accordingly.

(end of reply)

Reviewer 2:

The authors analyzed existence of furin cleavage sites at S1/S2 and S2' in coronaviruses related to human diseases. Especially, viruses from bats (the main hosts of coronaviruses) and intermediate hosts are included. This provides a more complete picture of the origin of furin cleavages sites in SARs-CoV-2. The study helps us to understand the evolution of viruses causing human diseases.

Reply: We thank you for reviewing our paper and your positive assessment of our work.

Reviewer 3:

The authors re-assessed diverse SrC from European horseshoe bats (samples were collected during 2008-2009) and showed that these bat SrC are likely to evolve a furin cleavage site through three possible mechanisms: recombination, multiple nucleotide insertions and nucleotide substitutions. The latter two mechanisms can be facilitated by a stem-loop secondary RNA structure enclosing the furin cleavage site and a high purine content in the external loop structure which are exactly the cases in some European bat-associated SrC (such as It1, SLO57 and BB99-04). In addition, SLO57 has the palindromic sequence CAGAC in the S1/S2 region which suggests that this genomic region may serve as an RNA signal for a recombination breakpoint.

In summary, I think this concise paper offered new evidences that support existing three mechanisms that favor a natural gain of the furin cleavage site in SARS-CoV-2.

The paper is well written and the figure is really nice. I fully support its publication without change.

Reply: We thank you for reviewing our paper and your positive assessment of our work.

Reviewer 4:

I think this is an interesting analysis of the potential route that SARS-CoV-2 could have used to acquire the furin cleavage site (FCS). My main question is why? That's a tricky one to answer for sure but I think the authors should address this issue in their discussion. What evolutionary pressure is driving acquisition of the FCS? Allied to this they should discuss how this FCS is lost in cell culture (i.e. on passage in VeroE6 cells). I think this will round off a useful analysis.

Reply: We thank you for reviewing our paper and your positive assessment of our work. In reply to your comments, we show that coronaviruses harboring a FCS may exist within the reservoir gene pool and hypothesize in the results and discussion section (see text blocks above) that an FCS may be fixed / selected after switching hosts in analogy to the transition from low-pathogenic to high-pathogenic avian influenza viruses. We are not certain that loss of the FCS in cell culture should be mentioned here, since those immortalized cell lines may not be representative of a hypothetical FCS turnover in nature. No text changes were thus made in reply to your comment on cell cultures.

REVIEWERS' COMMENTS:

Reviewer #4 (Remarks to the Author):

I am happy with the response of the Authors